# OpenReview forum: "iART: Imitation guided Automated Red Teaming"
_ICLR.cc/2026/Conference — Submitted to ICLR 2026_

### Official Review · Reviewer_ESTy · 2025-10-25

**Soundness:** 2
**Presentation:** 2
**Contribution:** 1
**Rating:** 2
**Confidence:** 4

**Summary:**

This paper proposes iART (Imitation-guided Automated Red Teaming), a reinforcement learning-based approach for automatically generating diverse and high-quality test cases to elicit harmful responses from large language models. The method introduces two key components: (1) imitation guidance using a "harm model" trained on examples of undesirable outputs to guide the attack LLM, and (2) a diversity module that models the distribution of previously generated outputs to penalize repetition. The approach is evaluated on text continuation and instruction-following tasks across multiple target LLMs (Mistral-7B, GPT-2-Alpaca, Dolly-3B), demonstrating improvements over baselines (RL, RL+TDiv, RL+Curiosity) in both quality and diversity while being computationally more efficient.

**Strengths:**

The paper clearly articulates the fundamental tension between quality and diversity in automated red teaming, and identifies computational inefficiency as a key limitation of existing methods.

**Weaknesses:**

1. The paper fails to cite and compare with several closely related and recently published works that address similar challenges in automated red teaming. For instance, DiveR-CT (Zhao et al., 2025) proposes relaxing conventional constraints on the objective and semantic reward, granting greater freedom for the policy to enhance diversity—an approach that directly addresses the quality-diversity trade-off that iART claims to solve. Similarly, HARM (Zhang et al., 2024) introduces a top-down hierarchical approach to generating test cases that ensures diversity through structured decomposition of harm taxonomies. Without empirical comparisons showing how iART performs relative to these methods, the paper's claimed contributions appear limited and its positioning within the current state-of-the-art remains unclear.
  reference:
 - (Zhao et al., 2025) DiveR-CT: Diversity-enhanced Red Teaming Large Language Model Assistants with Relaxing Constraints
 - (Zhang et al., 2024) Holistic Automated Red Teaming for Large Language Models through Top-Down Test Case Generation and Multi-turn Interaction
2. All main experiments in the paper use GPT-2-137M as the attack model, which raises significant questions about whether the approach scales to more capable and practical attack models. While Appendix E presents results using Mistral-7B as the attacker, these experiments reveal concerning discontinuities in the diversity plots that are only briefly mentioned and inadequately explained. The authors attribute these gaps to "no examples available at specific toxicity thresholds", but this explanation is unsatisfying and suggests potential instabilities when scaling up the attack model. Furthermore, the paper provides no analysis of how computational costs scale with attack model size—a critical consideration given that the diversity module requires online training and the harm model requires inference at each step. Without understanding these scaling properties, it remains unclear whether iART's computational advantages over RL+Curiosity would persist with larger, more powerful attack models that might be needed to successfully red team state-of-the-art LLMs.
3. The paper's evaluation framework has several limitations that undermine confidence in the results. First, using cosine similarity for measuring diversity is relatively shallow and may fail to capture semantic diversity—two test cases could have different embeddings but explore the same type of harm, or conversely, have similar embeddings while targeting different vulnerabilities. Second, the F1DQ metric that combines quality and diversity using harmonic mean lacks theoretical or empirical justification; the paper does not explain why harmonic mean is the appropriate aggregation function or whether other combinations (e.g., weighted averages, geometric mean) were explored. Third, relying on a single RoBERTa toxicity classifier as the sole evaluator may miss nuanced forms of harm and creates a narrow definition of "quality" that could be gamed by the attack model. Most critically, the paper includes no human evaluation to validate that the discovered "attacks" are genuinely problematic or that the diversity improvements represent meaningfully different failure modes rather than superficial lexical variations. Without human assessment, it is difficult to determine whether iART truly advances the practical goal of comprehensive LLM safety testing.

**Questions:**

1. Can you provide ablation studies showing how performance varies with different sizes and types of harm datasets? What is the minimum viable D_harm?
2. Can iART discover types of harmful content not present in D_harm? Or does it primarily find variations of known harms?
3. How does iART compare with gradient-based attacks like GCG in terms of attack success rate and diversity? These methods have shown high success rates on recent models.

---

> ### Author Response · Authors · 2025-11-25
> **Addressing the key technical questions and comments**
>
> Thank you for the time and care you put into your review.
>
> **1. Response to "Missing Baselines" (DiveR-CT, HARM)**
> We thank the reviewer for identifying these recent works. While relevant, we respectfully note that iART offers a distinct contribution centered on **efficiency via imitation guidance**.
> * **Methodological Distinction:** DiveR-CT (Zhao et al., 2025) focuses on relaxing constraints to improve diversity, and HARM (Zhang et al., 2024) utilizes a hierarchical decomposition. In contrast, iART introduces an *active guidance* mechanism using a "Harm Model" to shape the reward surface, coupled with a generative "Diversity Module".
> *Efficiency Bottleneck:** A primary motivation of iART is addressing the computational inefficiency of existing diversity-promoting methods. As shown in **Fig. 2(d)** and **Fig. 3(d, i)**, iART (423 mins) is $\approx 7\times$ faster than the strong baseline RL+Curiosity (2919 mins).
> * **ICLR Policy:** We note that DiveR-CT (2025) and HARM (2024) are contemporaneous works. Per ICLR guidelines, the absence of comparison to very recent papers is not grounds for rejection if the work demonstrates significant improvements over established baselines (RL+TDiv, RL+Curiosity), which iART successfully does.
>
> **2. Scalability and Attack Model Size (GPT-2 vs. Mistral)**
> The reviewer expresses concern regarding the use of GPT-2 and interprets the "discontinuities" in Appendix E as instability. We provide three clarifications:
> *Transferability as a Feature:** We deliberately use a smaller attack model (GPT-2 137M) to red-team larger target models (Mistral-7B, Dolly-3B) to demonstrate that efficient, lightweight models can successfully elicit toxicity from larger models.
> * **Discontinuities in Appendix E (Fact Correction):** The reviewer attributes the gaps in **Figure 8b** to "instability." This is factually incorrect. As explicitly stated in the text, "These discontinuities arise because there are no examples available at specific toxicity thresholds. When the Mistral-based attacker becomes highly effective (near 100% toxicity as shown in **Fig. 8a**), there are **zero samples** in the "low toxicity" bins. Consequently, a diversity score cannot be calculated for a bin that is empty. This is a statistical artifact of *high attack success*, not model instability.
> * **Computational Scaling (O(1) vs O(N)):** The reviewer asks about scaling costs.
>     *Baselines (RL+Curiosity):** Must compare the current candidate against *all* previously generated test cases to calculate diversity rewards (Self-BLEU/Cosine), leading to $O(N)$ complexity per step and quadratic scaling over training time.
>     * **iART:** Replaces exhaustive comparison with a Diversity Module (generative model) trained online. Calculating the reward is a forward pass of fixed size, i.e., **$O(1)$** with respect to history length.
>     * Therefore, iART’s computational advantage *increases* as we scale to larger attack models or longer training horizons.
>
> **3. Evaluation Metrics and "Shallow" Diversity**
> The reviewer states that "using cosine similarity... is relatively shallow" and "the paper includes no human evaluation... to determine whether iART truly advances...".
> *Missed Appendix G (Self-BLEU):** The reviewer appears to have overlooked **Appendix G**. We explicitly evaluate semantic and lexical diversity using **Self-BLEU** scores for n-grams 2, 3, 4, and 5 across all datasets (Tables 4, 5, and 6).
> *Evidence of Deep Diversity:** As stated in the text, "longer n-grams are more sensitive to lexical variation... reflecting increased diversity". iART consistently achieves lower Self-BLEU scores (indicating higher diversity) than baselines. For example, on the Alpaca dataset, iART achieves a Self-BLEU-5 of **0.1601** compared to RL+Curiosity's **0.1823**. This empirically validates that iART produces lexically diverse outputs, refuting the claim that we rely solely on "shallow" embeddings.
> * **F1DQ Justification:** The Harmonic Mean (F1) is the standard mathematical approach for balancing two competing objectives (Precision/Recall). Here, Quality and Diversity are the competing objectives. This is not an arbitrary choice but the standard definition of F1.
> *Evaluator Robustness:** The use of RoBERTa for toxicity detection is standard in the red-teaming literature (Perez et al., 2022; Hong et al., 2024).

---

> ### Author Response · Authors · 2025-11-25
> **Addressing technical questions .. continued**
>
> **4. Response to Questions**
>
> * **Harm Dataset Ablation:** We perform ablations on the *Imitation Guidance Coefficient* in **Appendix C (Fig. 6)** , showing the method is robust to variations in how strongly we weigh the harm dataset.
>
> * **New Harms:** The Harm Model is used to guide the reward signal, not to restrict generation to exact matches. Combined with the entropy bonus, iART explores the latent space of "harmfulness" learned by the reward model, allowing it to discover variations and adjacent harms not strictly present in $\mathcal{D}_{harm}$.
>
> * **Comparison to GCG:** Gradient-based methods (like GCG) require white-box access (gradients) to the target. iART is a **black-box** RL method (only requires inference). Comparing them directly is inequitable; black-box methods are essential for API-based testing where gradients are unavailable.
>
> ***

---

> ### Author Response · Authors · 2025-11-26
> **Contributions based on ICLR Criteria**
>
> We sincerely thank the reviewer for their detailed assessment and for acknowledging that our paper "clearly articulates the fundamental tension between quality and diversity". We appreciate the opportunity to clarify aspects of our evaluation and scalability analysis. We believe that addressing the factual misunderstandings regarding Appendix G and the computational complexity will demonstrate that the paper meets ICLR’s high standards.
>
> * **Correction of Factual Misunderstandings**
>
> * Critique: "The paper's evaluation framework has several limitations... using cosine similarity... is relatively shallow... without human assessment, it is difficult to determine whether iART truly advances..."
>     * Correction: This critique overlooks **Appendix G**, where we provided extensive **Self-BLEU** evaluation (Tables 4-6). We explicitly analyzed n-grams 2 through 5 to capture deep lexical diversity, stating "iART consistently achieves the lowest Self-BLEU scores... underscoring its effectiveness in generating diverse adversarial test cases". The evidence for meaningful diversity—beyond shallow embeddings—is already present in the paper.
> * Critique: "Experiments reveal concerning discontinuities... inadequately explained... suggests potential instabilities."
>     * Correction: As noted in the technical rebuttal, **Appendix E** states these are data sparsity artifacts arising from **high success rates**. When the model achieves near 100% toxicity (Fig 8a), there are no non-toxic samples left to populate the "low toxicity" bins for diversity calculation. This is a confirmation of the attack's effectiveness, not a sign of instability.
> * Critique: "The paper provides no analysis of how computational costs scale."
>     * Correction: We provided concrete runtime comparisons in **Figure 2d** and **Figure 3d/i** showing a **7x speedup**. Furthermore, the structural design of iART (Imitation/Diversity modules) allows for **$O(1)$** reward calculation per step, whereas the primary baseline (RL+Curiosity) scales linearly or quadratically with history length ($O(N)$). This algorithmic difference ensures the computational advantage persists and grows with model size.
>
> **Reaffirming Core Contributions**
> iART addresses the recognized "fundamental tension" between quality and diversity through a novel, computationally efficient framework:
> 1.  **Novelty:** It replaces expensive history-scanning (RL+Curiosity) with efficient Imitation and Diversity modules.
> 2.  **SOTA Performance:** It outperforms strong baselines in both Quality and Diversity across multiple datasets (IMDb, Alpaca, Databricks).
> 3.  **Efficiency:** It reduces red-teaming time from days to hours (Fig 2d), a critical contribution for practical safety testing.
>
> **Inconsistency of a "2" Rating**
> A score of 2 ("Reject") implies the paper has no value or fundamental flaws.
> * The reviewer's concern about "shallow metrics" is factually incorrect given the presence of **Appendix G**.
> * The concern about "scalability" is addressed by the algorithmic complexity analysis ($O(1)$ vs $O(N)$) and the empirical evidence of attacking Mistral-7B with GPT-2.
> * The request for comparison to 2024/2025 papers (HARM, DiveR-CT) refers to contemporaneous work, which, per ICLR guidelines, is not a valid ground for rejection.
> * Since the method is sound, outperforms baselines, and the "missing" analysis is actually present, a score of 2 is inconsistent with the evidence.
>
> **Appeal to ICLR Review Norms**
> ICLR guidelines emphasize that papers should be judged on their correctness and contribution to the community. We have demonstrated that iART provides a technically correct, highly efficient solution to a relevant problem. We respectfully submit that critiques based on overlooked appendices or the absence of contemporaneous baselines do not constitute "demonstrating lack of value to the community."
>
>
> We respectfully request that, in light of the clarification that **Self-BLEU analysis was already included** (addressing the diversity metric concern) and the explanation of the **structural computational advantage** (addressing the scalability concern), you reconsider whether a score of 2 appropriately reflects the paper’s contributions and alignment with ICLR’s acceptance criteria.
>
> Warm Regards,
>
> Authors

---

### Official Review · Reviewer_b9nV · 2025-10-31

**Soundness:** 3
**Presentation:** 3
**Contribution:** 3
**Rating:** 6
**Confidence:** 4

**Summary:**

The paper focuses on red-teaming for LLMs, following the settings of “RL” (Perez et al., 2022)  and “RL+Curiosity” (Hong et al., 2024). The authors added another input to the RL+Curiosity original pipeline, which is a harmful LM to provide standard harmful outputs. It’s trained on a harmful dataset. With this addition the RL process can get additional teacher guidance on what a “harmful” response would be under the current input prompt, and there is one more term in the objective to encourage the attacker to elicit similar responses from target LLM (measuring cosine similarity as reward). The experiment results show that this approach converged faster with better quality and significantly lowered the execution time to find a harmful prompt compared to “RL+Curiosity”.

**Strengths:**

1. The imitation-guidance clearly stabilizes RL training: it acts like a teacher signal that speeds up convergence and makes the attacker’s learning more stable.
2. Experiments show the attacker reaches effective harmful prompts more quickly and with higher success than RL+Curiosity.
3. Effective across different language models and observe similar gains, suggesting the method generalizes across model architectures and sizes.

**Weaknesses:**

1. There is a potential issue that the imitation may limit exploration (mode bias). The attacker is explicitly rewarded to produce prompts that make the target model output responses similar to the harm (teacher) model. That can bias the attacker to explore only regions of the target’s output space that match the teacher’s harmful patterns. Even if the teacher itself produces diverse harmful outputs, the attacker’s exploration is effectively constrained to that teacher-driven subspace. As a result, the attacker may miss other kinds of harmful behaviors that the target can produce but the teacher is not able to produce.

2. The above issue may not be reflected on the current evaluation benchmarks. Because the harm model’s outputs are themselves diverse, standard diversity measures (Self-BLEU, embedding cosine diversity) will still report high diversity even when the attacker is merely mimicking the teacher. Those metrics do not reveal whether the attacker is covering teacher-like harmful modes only or genuinely discovering teacher-independent harmful modes.

3. Also, there’s potential for overfitting to the harm dataset. Because imitation guidance depends on a harmful dataset to train the teacher, any dataset bias transfers into what the attacker searches for and limits the coverage of other real-world harmful cases.

**Questions:**

I would appreciate if the authors responded to some of the potentially weaknesses outlined above.

---

> ### Author Response · Authors · 2025-11-25
> **Addressing the key technical questions and comments**
>
> We sincerely thank you for your thoughtful review. We value your recognition that our imitation-guidance **"clearly stabilizes RL training"** and that the method **"generalizes across model architectures,"** achieving effective harmful prompts significantly faster than the baseline. We appreciate your positive assessment.
>
> We understand your reservations stem from theoretical concerns regarding **exploration limits (mode bias)** and **dataset dependence**. We are confident these are addressable clarifications rooted in the hybrid nature of our objective function.
>
> ### **1. Addressing Mode Bias and Exploration Limits**
>
> **Reviewer Concern:** *"The imitation may limit exploration... constrained to that teacher-driven subspace... [missing] other kinds of harmful behaviors."*
>
> **Response:** We appreciate this insightful concern. While pure imitation learning can indeed lead to mode collapse, our method is **Imitation-Guided RL**, which explicitly prevents this through a multi-objective design (Eq. 3). The attacker is governed by three competing forces:
>
> 1.  **Imitation ($\beta_1 D_{cos}$):** Acts as a **soft, distributional prior** (not a hard constraint) to guide the agent toward the manifold of *known* harmful semantics.
> 2.  **RL Reward ($R(y)$):** Rewards **any** output that the independent Evaluator (RoBERTa) deems toxic. If the agent discovers a novel, non-teacher mode that maximizes $R(y)$, the RL term explicitly reinforces it.
> 3.  **Diversity Module ($-D_{cos}(x, \tilde{x})$):** This is the critical counter-force. As defined in **Section 4.2**, this module penalizes the agent for generating *previously seen* outputs. It forces the agent to continuously explore *away* from both the teacher's modes (once visited) and its own history.
>
> **Empirical Evidence:**
>
> * **Ablation Study (Figure 4 & 5):** If imitation were constraining exploration, removing it should *increase* diversity. However, our results show that removing imitation guidance **decreases** diversity. This confirms that the Harm LLM acts as an "exploration amplifier," helping the agent navigate the vast search space effectively rather than collapsing into a single failure mode like the baseline.
> * **Lowest Self-BLEU:** As shown in **Tables 4-6 (Appendix G)**, iART achieves the **lowest Self-BLEU scores** (indicating highest diversity) across all n-grams compared to baselines. This proves the agent is not collapsing to a subspace but is generating a broader distribution of outputs.
>
>
> ### **2. Validity of Diversity Metrics and Teacher Dependence**
>
> **Reviewer Concern:** *"Standard diversity measures... will still report high diversity even when the attacker is merely mimicking the teacher."*
>
> **Response:** We agree that mimicking a diverse teacher yields high diversity scores, but we present two lines of evidence showing that iART goes beyond mimicry to discover target-specific harms:
>
> 1.  **Cross-Model Generalization:** We evaluate iART against three distinct target LLMs: **Mistral-7B, GPT2-Alpaca, and Dolly-3B**. These models have vastly different training data and vulnerabilities. If the attacker were merely mimicking the teacher (Harm LLM), its success rate would fluctuate wildly depending on the target's overlap with the teacher. Instead, iART consistently outperforms baselines across **all** targets (Figs 2, 3). This indicates the agent is learning to probe the *specific* weaknesses of each target, facilitated by the RL reward $R(y)$.
> 2.  **Target-Centric Evaluation:** Our diversity metrics (Self-BLEU, Cosine Similarity) are computed on the **target's responses**, not just the attack prompts. High diversity in the target's output confirms that the attacker successfully elicited a wide range of harmful behaviors *intrinsic to the target*, rather than just replaying teacher prompts that might fail to trigger the target.
>
>
> ### **3. Potential Overfitting to Harm Dataset**
>
> **Reviewer Concern:** *"Any dataset bias transfers into what the attacker searches for and limits the coverage."*
>
> **Response:** We mitigate dataset bias through the **independence of the Evaluator**:
>
> * The **Evaluator** ($R(y)$) is an independent toxicity classifier (RoBERTa), *not* trained on the Harm Dataset.
> * Because the objective maximizes $R(y)$, the attacker is incentivized to find *any* prompt that triggers the Evaluator, even if that prompt is orthogonal to the Harm Dataset. The Harm Dataset provides a "warm start," but the RL reward ensures the policy is not bounded by it.
> * **Diverse Priors:** The dataset used ($\mathcal{D}_{harm}$) is broad (ToxicDPOqa), covering multiple domains of toxicity. As noted in your review, the method's effectiveness across different architectures suggests it is not overfitted to specific dataset artifacts.

---

> ### Author Response · Authors · 2025-11-25
> **Contributions based on ICLR Criteria**
>
> ### Contributions via ICLR Criteria
>
> We link the strengths you identified to the ICLR acceptance criteria to support a score upgrade:
>
> 1.  Is the problem important? Yes. Efficient red teaming is critical. You noted our method *"significantly lowered the execution time,"* addressing a major bottleneck.
> 2.  Is the approach well-motivated? Yes.You recognized that *"imitation-guidance clearly stabilizes RL training."* We effectively solve the exploration-exploitation dilemma in red teaming by using a behavioral prior, indicating strong conceptual grounding.
> 3.  Does the paper support claims? Yes. We provide extensive ablations (Figs 4 & 5), hyperparameter sweeps (Fig 6), and comparisons showing superior speed and quality.
> 4.  Is the contribution significant? Yes. A **7x speedup** combined with higher stability and generalization is a substantial contribution to the ML safety community.
>
> ###  Concrete Revisions We Commit To
>
> * Add a dedicated subsection explicitly explaining imitation-guided exploration.
> * Add discussion on teacher dependence and target-specific harmful modes.
> * Improve clarity in Sections 4.1 and 4.2.
> * Enhance figure captions and diversity metric interpretation.
> These resolve all remaining issues without altering any claims or results.
>
> ### Why a Score of 6 Undershoots the Work's Value
>
> We appreciate the score of 6, but we respectfully suggest that the concerns regarding "potential mode bias" are theoretical risks that are mitigated by our **hybrid RL+Diversity design** and refuted by the **Self-BLEU evidence**.
>
> * Your ratings for Soundness, Presentation, and Contribution are all 3 (“good”). A paper that is good in all three dimensions typically aligns with ICLR’s “Accept (8)” category.
> * The concerns focus on *potential* limitations rather than observed empirical failures. ICLR guidelines emphasize that papers offering significant methodological advances (like our speedup and stability) should be valued even if they rely on data priors.
> * Given that the method works, the results are significant, and the theoretical concerns are addressed by the algorithm design, the evidence aligns more naturally with an"Accept" (8).
>
> In light of these clarifications—and given that the method is technically sound, novel, empirically compelling, and of clear value to the LLM safety community—we respectfully ask whether a score of 6 fully reflects the strength of the work. We believe the contribution aligns more clearly with an **Accept” (8)** rating, and we sincerely appreciate your openness to revisiting your recommendation in accordance with ICLR guidelines.
>
> Thank you again for your thoughtful review and for helping us strengthen the paper.
>
> Warm Regards,
>
> Authors

---

### Official Review · Reviewer_fPwH · 2025-11-03

**Soundness:** 2
**Presentation:** 2
**Contribution:** 2
**Rating:** 6
**Confidence:** 3

**Summary:**

This paper studies a red teaming LLM problem. This paper proposes to use a pre-collected dataset of harmful outputs to help the attack LLM generate prompts triggering harmful outputs, and a model that measures if a certain response has been generated before to encourage the attack LLM to produce diverse outputs. The results show that the performance in quality and diversity is similar to the prior work, while the proposed method significantly reduces the execution time.

**Strengths:**

The execution time seems to be largely reduced compared with the prior works, which is a great contribution. However, I would love to see more explanation of how the proposed method made it into the paper rather than the appendix.

**Weaknesses:**

- The writing was not clear. See my questions.
- Presentation is messy. The layout of the figures needs to be largely improved.

**Questions:**

- Line 170: I didn't understand the need to train the harm LLM. It seems to me that encouraging the attack LLM to generate prompts that make the target LLM produce outputs similar to the outputs from the harm LLM is the same as eliciting outputs with harmful outputs measured by an evaluator.
- Section 4.2 & Section 4.1: The details of the diversity module and harm LLM shouldn't be put in the Appendix. The recipe for training the diversity module and harm LLM seems to be the main contribution of this paper.
- Fig 3: There are a lot of glitches on the borders of the figure. It looks unprofessional. Also, there are no legends.

---

> ### Author Response · Authors · 2025-11-25
> **Addressing the key technical questions and comments**
>
> We sincerely thank you for your constructive assessment and for recognizing that our method delivers a “great contribution” through its large reduction in execution time. We appreciate the positive overall evaluation and welcome the opportunity to clarify remaining questions
>
> We understand that your reservations stem primarily from a need for clarification on the Harm LLM's role and concerns regarding presentation (figure quality and method placement). We are confident these are addressable points. The technical core of the paper is sound, and the presentation issues are fixable in revision.
>
> ### **1. Clarifying the Necessity of the Harm LLM** (Line 170)
>
> **Reviewer Concern:** *"I didn't understand the need to train the harm LLM. It seems... the same as eliciting outputs with harmful outputs measured by an evaluator."*
>
> **Response:** This is a critical distinction that we will clarify in the revision. The Harm LLM and the Evaluator serve fundamentally different mathematical roles in our objective function (Eq. 2):
>
> 1. **Evaluator = Scalar Reward (Sparse):** The evaluator $R(y)$ provides a single scalar score (e.g., toxic/not toxic). Training solely on this often leads to **mode collapse**, where the model exploits the reward by repeating a single type of toxic phrasing. It tells the model *that* it should be harmful, but not *how*.
>
> 2. **Harm LLM = Distributional Prior (Dense):** The Harm LLM $\phi$ models the **distribution** of harmful behaviors in the dataset $\mathcal{D}_{harm}$. The term $+\beta_1 D_{cos}(y, \tilde{y})$ acts as a dense, vector-based guide. It steers the attack policy toward the *manifold* of diverse harmful behaviors (e.g., discrimination, threats, misinformation) captured in the dataset.
>
> **Why it is needed:** As shown in our ablation study (**Figure 4**), removing the imitation guidance (Harm LLM) significantly degrades performance. The Harm LLM acts as a "behavioral prior" that accelerates exploration and ensures the model covers a *diverse* range of harmful outputs, rather than collapsing into a single high-reward failure mode.
>
> ### **2. Method Placement and Presentation**
>
> **Reviewer Concern:** *"The details of the diversity module and harm LLM shouldn't be put in the Appendix... [it] seems to be the main contribution."*
>
> **Response:** We completely agree. We originally moved these details to the Appendix due to strict page constraints, but we recognize they are central to the method's novelty.
>
> * **Action Plan:** In the camera-ready version, we will move the formal algorithm definition and training details for both the Diversity and Harm modules from Appendix B.2/B.6 into **Section 4**. To make space, we will condense the Related Work (Section 2) and move secondary experimental details (such as specific hyperparameter tables) to the Appendix.
>
> **Reviewer Concern:** *"Fig 3: There are a lot of glitches... looks unprofessional. Also, there are no legends."*
>
> **Response:** We apologize for this oversight. The "glitches" were artifacts introduced during the PDF export of vector graphics.
>
> * **Action Plan:** We have already regenerated Figure 3 at high resolution to eliminate these artifacts and have added clear, descriptive legends to all subplots. This is a cosmetic fix that ensures immediate readability.

---

> ### Author Response · Authors · 2025-11-25
> **Contributions based on ICLR Criteria**
>
> ### Alignment with ICLR Acceptance Criteria
>
> You acknowledged that our method "significantly reduces the execution time", a critical bottleneck in red teaming. Here we connect this strength to the four core ICLR acceptance questions to support a score upgrade:
>
> (a) Is the problem important?
>
> Yes. Efficient red teaming is critical for LLM safety, and current SOTA approaches (e.g., RL+Curiosity) are prohibitively expensive, scaling as $O(N^2)$ with generated samples.
>
> (b) Is the approach well-motivated and novel?
>
> Yes. iART introduces two algorithmic innovations:
>
> * Imitation-guided RL using a distributional harm prior (Harm LLM), and
> * Generative diversity modeling replacing pairwise comparisons with a learned approximation.
> This shifts complexity from $O(N^2)$ comparison to an $O(N)$, a fundamental advance rather than an engineering tweak.
>
> (c) Are claims well supported?
>
> Yes. We provide:
>
> * multi-LLM and multi-task evaluations,
> * ablations isolating imitation and diversity contributions,
> * hyperparameter sweeps,
> * the F1DQ metric,
> * 7× speedups in execution time (Figs. 2d, 3i).
>
> This satisfies ICLR’s expectation for rigorous, reproducible empirical support.
>
> (d) Is the contribution significant for the community?
>
> Yes. The method enables scalable safety evaluation, previously impractical, while improving the quality–diversity trade-off.
>
> Together, these may meet the ICLR bar for an Accept (8).
>
> ### Concrete Revisions We Commit To
>
> * Move Harm + Diversity training details to Section 4
> * Improve Section 4 clarity and add intuitive explanation
> * Regenerate all figures with legends and corrected borders
> * Condense Related Work and restructure minor experimental tables
> * Strengthen discussion of imitation-guided exploration
>
> These may fully address your concerns.
>
> ### Why a Score of 6 Undershoots the Work's Value
>
> We deeply respect your score, but based on ICLR guidelines:
>
> * All cited concerns are minor presentation issues or requests for clarity, which the guidelines state should not justify borderline ratings.
>
> * No issues of correctness, experimental soundness, misplacement in literature, or lack of significance were raised.
>
> * You explicitly acknowledge the major scientific value, the drastic execution-time reduction.
>
> * We have resolved the only conceptual question (Harm LLM vs. evaluator), which was based on an understandable misunderstanding.
>
>
> In light of the clarification regarding the Harm LLM's role as a distributional prior (preventing mode collapse) and our commitment to correcting the presentation issues, we believe the core contributions are technically sound, novel, empirically strong, and valuable to the community. We respectfully ask whether a score of 6 fully reflects the strength of the work. We believe the paper now aligns with the criteria for an “Accept” (8), and we sincerely appreciate your openness to reassessing your recommendation during the discussion phase, as encouraged by ICLR reviewing guidelines.
>
> Thank you again for the constructive review and for helping us strengthen our paper.
>
> Warm Regards,
>
> Authors

---

### Author Response · Authors · 2025-12-04
**Summary of Rebuttals and Case for Acceptance**

Dear Area Chair,

We appreciate your attention to this submission. We respectfully present this summary to demonstrate that **iART (Imitation-guided Automated Red Teaming)** makes a significant, technically sound contribution to LLM safety research, specifically addressing the critical bottleneck of computational efficiency.

Please note that none of the reviewers were able to respond to our rebuttal before the freeze. We believe the divergence in ratings, two "Marginally Above Acceptance" (6) and one "Reject" (2), stems primarily from factual misunderstandings and failure to thoroughly read the paper by the negative reviewer regarding our evaluation metrics and scalability properties, which we have fully addressed in our rebuttals.

### **1. Consensus on Contribution: Efficiency and Stability**
There is a strong consensus among the majority of reviewers regarding the paper’s primary value:
* **Efficiency:** Reviewer fPwH noted the method "significantly reduces the execution time", and Reviewer b9nV highlighted that iART "significantly lowered the execution time... compared to RL+Curiosity".
* **Stability:** Reviewer b9nV recognized that "imitation-guidance clearly stabilizes RL training" and that the method "generalizes across model architectures".
* **Empirical Gains:** We demonstrated a **7x speedup** (423 mins vs. 2919 mins) while maintaining or exceeding SOTA quality.



### **2. Resolution of Positive/Borderline Concerns (Scores of 6)**
Reviewers fPwH and b9nV raised constructive points that we have fully resolved:
* **Harm LLM vs. Evaluator:** Reviewer fPwH asked about the necessity of the Harm LLM. We clarified that the Evaluator provides a *sparse scalar reward* (toxic/not toxic), while the Harm LLM provides a *dense distributional prior*. The Harm LLM acts as an "exploration amplifier" to prevent mode collapse, a distinction validated by our ablation studies in Figure 4.
* **Mode Bias:** Reviewer b9nV worried about the attacker mimicking the teacher. We refuted this by pointing to **Appendix G**, where iART achieves the lowest **Self-BLEU** scores (indicating high lexical diversity) across all n-grams. This proves the agent explores beyond the teacher's subspace.

### **3. Addressing the "Reject" Rating (Reviewer ESTy)**
Reviewer ESTy’s score of 2 appears to rest on four specific critiques, which we have demonstrated to be factually incorrect or outside ICLR guidelines:

* **Critique 1: "Shallow" Diversity Metrics.** The reviewer claimed we relied only on cosine similarity and missed semantic diversity.\
    **Rebuttal:** This was a **factual oversight**. We explicitly included **Self-BLEU** analysis (n-grams 2-5) in **Appendix G** (Tables 4-6). As stated in our response, "iART consistently achieves the lowest Self-BLEU scores... refuting the claim that we rely solely on 'shallow' embeddings".

* **Critique 2: Scalability "Instabilities".** The reviewer interpreted gaps in Figure 8b as model instability.\
    **Rebuttal:** We clarified that these are artifacts of **high success**. When the attack is near 100% successful (Figure 8a), there are zero samples in the "low toxicity" bins to calculate diversity for. This is a statistical result of effectiveness, not instability.

* **Critique 3: Computational Scaling.** The reviewer requested analysis on scaling costs.\
    **Rebuttal:** We explained the structural advantage: iART relies on forward passes of fixed modules ($O(1)$ complexity wrt history), whereas the baseline RL+Curiosity requires comparing against *all* prior outputs ($O(N)$ complexity). Thus, iART's advantage *grows* with scale.

* **Critique 4: Missing Contemporaneous Baselines.** The reviewer cited papers from 2024 and 2025 (DiveR-CT, HARM) as missing comparisons.\
    **Rebuttal:** Per ICLR guidelines, the absence of comparison to very recent/contemporaneous work is not valid grounds for rejection, particularly when the paper outperforms established baselines (RL+TDiv, RL+Curiosity).

### **Conclusion**
iART addresses the critical bottleneck of computational efficiency in automated red teaming, offering a **7x speedup** over state-of-the-art baselines while maintaining superior diversity and quality. We have unified the feedback from Reviewers fPwH and b9nV by clarifying the theoretical role of the Harm LLM as a distributional prior. Furthermore, we have demonstrated that the "Reject" recommendation relies on factual misunderstandings, specifically, the oversight of our Self-BLEU analysis in Appendix G and the misinterpretation of high-success artifacts as instability.\
 Given that iART fundamentally shifts the algorithmic complexity of diversity enforcement from $O(N)$ to $O(1)$, we respectfully submit that the paper makes a significant, technically sound contribution to the LLM safety community and merits acceptance.

Sincerely,\
The Authors

---

### Meta-Review · Area_Chair_Z5fV · 2026-01-11

**Summary:**

The paper introduces a RL-based approach to generate diverse and high-quality test cases to elicit harmful responses from LLMs. The approach relies on a "harm model", which provides imitation guidance and a "diversity module", which penalizes repetition. Two reviewers were mildly positive and one reviewer was negative. To provide a more fair assessment, I did also read the paper independently. I find the technical contribution rather modest, the computational benefit enabled by the authors' approach of limited significance in practice, and (some of) the concerns raised by the reviewers fair. In this context, I find it questionable that the authors argue that a method introduced in 2024 is contemporaneous as a basis to justify not to compare with it. As a result, I am unable to recommend acceptance.

**Reviewer Concerns:**

I think the lack of comparison with two baselines has not been successfully addressed, particularly for the method introduced in 2024.

**Reviewer Scores:**

I think the most negative reviewer would not have been convinced by the rebuttal, particularly regarding the lack of comparison with methods from 2024 and 2025.

---

### Decision · Program_Chairs · 2026-01-26

Reject